# Conversion to belatacept after lung transplantation: *Report of 10 cases*

**Olivier Brugière**[1]*, **Alexandre Vallée**[2], **Quentin Raimbourg**[3], **Marie-Noelle Peraldi**[4], **Sylvie Colin de Verdière**[1], **Laurence Beaumont**[1], **Abdulmonem Hamid**[1], **Mathilde Zrounba**[1], **Antoine Roux**[1], **Clément Picard**[1], **François Parquin**[5], **Matthieu Glorion**[6], **Julie Oniszczuk**[4], **Alexandre Hertig**[4], **Hervé Mal**[7], **Vincent Bunel**[7]

**1** Service de Transplantation Pulmonaire, Hôpital Foch, Suresnes, France, **2** Service de Biostatistique, Hôpital Foch, Suresnes, France, **3** Service de Néphrologie, Hôpital Bichat, Paris, France, **4** Service de Néphrologie, Hôpital Saint-Louis, Paris, France, **5** Service de Réanimation, Hôpital Foch, Suresnes, France, **6** Service de Chirurgie Thoracique, Hôpital Foch, Suresnes, France, **7** Service de Pneumologie B et de Transplantation Pulmonaire, Hôpital Bichat, Paris, France

* o.brugiere@hopital-foch.com

## Abstract

### Background

Calcineurin inhibitors (CNIs) remain the cornerstone of maintenance immunosuppression (IS) after lung transplantation (LTx), although CNI-related life-threatening toxic effects may occur. Belatacept, a novel immunosuppressant that blocks a T-cell co-stimulation pathway, is a non-nephrotoxic drug indicated as an alternative to CNIs in kidney Tx. In LTx, there are only a few reports of belatacept conversion as a CNI-free or CNI-sparing IS treatment.

### Methods

We reviewed a series of 10 LTx recipients with conversion to a CNI-free belatacept IS regimen within the first year post-LTx (n = 7) or a belatacept/low-dose CNI combination after the first year (n = 3).

### Results

Use of belatacept was triggered by severe renal failure in 9 patients and under-IS with previous other IS-related toxicities in 1 patient. Mean estimated glomerular filtration rate after starting belatacept significantly improved at 6 months after initiation and at the last-follow-up (p = 0.006, and p = 0.002 respectively). The incidence of recurrent and/or severe acute cellular rejection (ACR) episodes was high in patients with CNI-free belatacept-based IS (n = 4/7). Chronic graft allograft dysfunction developed in 2 of 9 recipients under belatacept IS. Belatacept was stopped in 6 patients because of recurrent/severe ACR (n = 3), recurrent opportunistic infections (n = 1), center modified policy (n = 1), or other cause (n = 1).

### Conclusion

Early conversion to CNI-free belatacept-based IS improved renal function in this series but was counterbalanced by a high incidence of recurrent ACR, including life-threatening

**Data Availability Statement:** All relevant data are within the paper and its Supporting information files.

**Funding:** The author(s) received no specific funding for this work.

**Competing interests:** The authors have declared that no competing interests exist.

episodes. Other studies are needed to better determine the indications for its use after LTx, possibly with lower immunological risk IS regimens, such as CNI-sparing belatacept.

## Introduction

Currently, calcineurin-inhibitors (CNIs) remain the main immunosuppressive (IS) therapy for long-term maintenance IS after lung transplantation (LTx). Nevertheless, severe CNI-related toxic effects may occur after LTx, mainly manifested by renal insufficiency [1–3], or neural toxicity [4]. These toxic effects may be life-threatening, and renal insufficiency has been found as an independent factor of morbi-mortality after LTx [1–3]. Hence, new strategies to prevent CNI toxic effects are needed, especially in case of rapidly progressive kidney insufficiency [5].

Belatacept, a new non-nephrotoxic immunosuppressive agent, is a selective costimulation blocker that binds CD80 and CD86, thereby blocking CD28-mediated costimulation in the T-cell activation cascade [6, 7]. In kidney-Tx recipients, it has shown promise as being an alternative agent by preserving renal function, and was approved as a first-line treatment in kidney transplantation to replace a CNI-based regimen [6, 7]. Additionally, kidney-Tx recipients receiving belatacept are less likely to develop donor-specific antibodies (DSAs) [7], which suggests another potential benefit of its use.

Only a few experiences with belatacept have been reported in lung Tx [8–14], and the risk/benefit balance is still not well assessed in this indication. Here we report our experience in a series of 10 patients who were switched to an IS-regimen including belatacept for CNI-related renal toxicity. Belatacept was administered as a CNI-free belatacept-based IS regimen or as a CNI-sparing belatacept IS regimen. We report the outcome of these patients during belatacept exposure, including kidney function outcome, and the occurrence of acute cellular rejection (ACR), chronic lung allograft dysfunction (CLAD), and opportunistic infections until the last follow-up.

## Methods

### Lung Tx recipient population switched to belatacept

Data for 10 LTx recipients from Bichat and Foch Tx centers (France) for whom belatacept was used were analyzed within the 2016–2021 period. Maintenance IS therapy was similar in both centers, including tacrolimus (C0 trough level = 8–12 ng/L), mycophenolate mofetil (2 g/day) and prednisone (5 mg/day). An induction therapy with thymoglobulins or basiliximab was used only at Foch. The protocols of IS regimen administered in the 2 centers have been previously reported [15], and are detailed in S1 Data [15].

The first 7 patients (Bichat center) were switched to a CNI-free belatacept-based IS regimen during the first year post-LTx, associated with mycofenolate mofetil [MMF] continuation. Within the first 3 months after belatacept initiation, systematic transbronchial biopsies (TBBx) were planned at 1, 2, and 3 months post-conversion at the Bichat center.

Three other patients followed at the Foch center were switched to a CNI-sparing belatacept IS regimen (targeting very-low dose CNI [target C0 tacrolimus levels = 2–3 ng/ml, or target ciclosporine C0 = 40–75 ng/ml]). The surveillance protocol for AR episodes and CLAD diagnosis in both centers was previously reported [15–18] (see S1 Data).

### Belatacept protocol

All LTx recipients were considered for a switch from CNI to belatacept in case of severe renal insufficiency, with a high probability of permanent dialysis in case of continuation of CNI

treatment. Belatacept was initiated after verifying positive antibody serostatus for Epstein–Barr virus (EBV), with a "less-intensive" regimen reported in kidney-Tx [6, 19] (see S1 Data). CNI was progressively tapered and then stopped on day 14 in 7 patients (Bichat center) or only tapered to a very-low dose in 3 patients (Foch center, T0 trough level = 2–3 ng/ml) (see S1 Data). Glomerular filtration rate (GFR) was estimated by using the Chronic Kidney Disease Epidemiology Collaboration equation for creatinine [20].

This study was approved by an institution ethics committee (IRB00012437), and conducted in accordance with good clinical practices and recommendations concerning human research contained in the Declaration of Helsinki. All patients gave their informed consent to be included in the study.

## Statistics

Continuous variables are described with mean (SD) or median (range) and were compared by Student $t$ test or Mann–Whitney U-test. Statistical significance was set at $p < 0.05$.

## Results

### Patients

Characteristics of patients are detailed in Table 1. For the 7 patients switched to a CNI-free belatacept-based IS regimen, belatacept was initiated at a median postoperative day (POD) of 112 (45–330). Causes of conversion to belatacept were acute, acute-on-chronic, or chronic renal insufficiency, including the following (Table 1): CNI toxic effects (n = 3), CNI toxic effects associated with acute tubular necrosis (n = 2), focal segmental glomerulosclerosis associated with CNI toxic effects (n = 1), and thrombotic microangiopathy (TMA) attributed to tacrolimus (n = 1). All patients had normal pre-LTx creatinine clearance, except for patient 4 with focal segmental glomerulosclerosis on renal biopsy diagnosed for 3 years and creatinine clearance 55 ml/min before LTx. Three patients had associated possible cofactors for renal insufficiency such as arterial hypertension or diabetes mellitus.

In the 3 other patients who received belatacept associated with CNI reduction doses (very low doses, tacrolimus C0 levels = 2–3 ng/L), causes for starting belatacept were severe renal insufficiency due to TMA (n = 2) or kidney insufficiency associated with under-IS in a patient with CLAD (patient 9, Table 1). Belatacept was initiated at a median POD of 635 (481–3800) in these 3 patients. Median exposure duration to belatacept for the 10 patients was 12 months (4.7–25).

### Renal function outcome after lung transplantation

Post-LTx outcome of renal function before conversion to belatacept for all patients is shown in Fig 1, as individual creatinine values of the 10 patients from day 0 of LTx to the day of conversion to belatacept. During this period, mean creatinine value increased from 66 ± 27 micromole/L at day 0 of LTx to 275 ± 116 micromole/L at the date of conversion to belatcept (p = 0.0002).

After starting belatacept, the 7 patients with CNI-free belatacept-based IS all exhibited a significant increase in estimated GFR (eGFR) values at 1 month, 3 months, and 6 months. Mean eGFR under belatacept increased from 21 ±9 mL/min (n = 10) to 43 ± 24 mL/min (n = 10) at 1 month (p = 0.01), to 50 ± 25 mL/min (n = 9) at 3 months (p = 0.005), to 55 ± 30 mL/min (n = 8) at 6 months (p = 0.006), and to 54 ±28 mL/min (n = 9) at the last-follow-up under belatacept (p = 0.002). One patient on dialysis was successfully weaned off renal replacement therapy (patient 1). The 3 last patients with CNI-sparing belatacept IS with very-low dose CNI

**Table 1. Characteristics of patients.**

| Pt no | Age | Sex | Initial disease | LTx type | CMV (D/R) HSV (R) | HTA Diabetes | Induction therapy | Cause for switch | Delay post-Tx before starting belatacept | CNI /IS Pre-switch | Associated-IS Regimen after switch to belatacept |
|---|---|---|---|---|---|---|---|---|---|---|---|
| 1 | 59 | M | COPD | BLT | CMV -/+ HSV+ | - | no at day 0. Basiliximab for CNI holiday (M3) | ATN + CNI toxicity | D 162 | FK MMF | MMF 0.5gx2/d prednisone |
| 2 | 46 | F | COPD | BLT | CMV -/-HSV+ | HTA | no | TMA $^\S$ | D 96 | FK MMF | MMF: 1gx2/d then evero > 3 mo prednisone |
| 3 | 49 | F | COPD | BLT | CMV +/-HSV+ | - | no at day 0. Basiliximab for CNI holiday (M1) | ATN + CNI toxicity | D 40 | FK MMF | MMF: 1gx2/d prednisone |
| 4 | 44 | F | Fibrosis | BLT | CMV +/+ HSV+ | HTA DM | no | ATN + FSG + CNI toxicity | D 45 | FK MMF | MMF: 0.5gx2/d prednisone |
| 5 | 57 | F | COPD | BLT | CMV -/+ HSV+ | - | no at day 0. Basiliximab for CNI holiday (M1) | Renal CNI toxicity | D 112 | FK MMF | MMF: 0.5gx2/d prednisone |
| 6 | 58 | M | Fibrosis | SLT | CMV -/+ HSV+ | HTA DM | no | Renal CNI toxicity | D 370 | FK/MMF | MMF: 1gx2/d prednisone |
| 7 | 65 | F | Fibrosis | SLT | CMV +/+ HSV+ | - | no | Renal CNI toxicity | D 330 | FK MMF | MMF: 1gx2/d prednisone |
| 8 | 68 | M | Fibrosis | SLT | CMV +/+ HSV+ | HTA | no | TMA | Y1M4 | FK Evero | FK very low + Evero very low prednisone |
| 9 | 63 | F | COPD | BLT | CMV +/+ HSV+ | HTA | Thymo | Renal CNI toxicity Under-IS* | Y10M5 | FK evero | FK very low prednisone |
| 10 | 59 | F | COPD | BLT | CMV-/+ HSV+ | HTA DM | Thymo | TMA CNI toxicity | Y1M9 | FK MMF | FK very low prednisone |

COPD: chronic obstructive pulmonary disease; SLT: single-lung transplantation; BLT, bilateral-lung transplantation; CMV D/R: Serology of cytomegalovirus from D (donor) and R (recipient); HSV, serology of herpes simplex virus from R (recipient); HTA, systemic hypertension; DM: diabetes mellitus; Thymo: Thymoglobulins as induction therapy. CNI: calcineurine inhibitors; CNI holiday: calcineurin inhibitor holiday was used in 3 patients, with administration of basiliximab 20 mg/d on day 0 and 4, associated with reduced CNI dosage (Tacrolimus C0 = 2–3 ng/ml for 10 days); ATN: acute tubular necrosis; TMA: thrombotic microangiopathy; CNI; calcineurin inhibitor; FSG: focal segmental glomerulosclerosis; Y: year; M: month; IS: Immunosuppressive; MMF: mycophenolate mofetil; FK: Tacrolimus; Evero: everolimus; Evero very low: everolimus C0 levels = 2–3 ng/ml.

*Under-IS: under-immunosuppression (after successive stopping of MMF, azathioprine, and everolimus for toxicities). Prednisone: prednisone low-dose (5 mg/d).

continuation also showed a significant increase in eGFR at the last follow-up under belatacept versus at belatacept initiation (35 ± 4 mL/min vs 17 ± 3 mL/min; p = 0.004). Outcome of individual creatinine clearance values for all patients under belatacept is shown in Fig 2A. Among all patients, mean creatinine value decreased from 274 ± 116 μmol at the date of conversion to belatcept to 126 ± 65 μmol at the last day of follow-up under belatacept (p = 0.0009).

During follow-up, belatacept was discontinued in 6 patients who were re-switched to a standard CNI-based IS. Causes of discontinuation were as follows: recurrent/severe ACR [n = 3], viral infection [n = 1], change in center policy [n = 1], and other [n = 1]). In the 5 patients with available follow-up after belatacept discontinuation, creatinine value was 154 ± 69 μmol /L at the date of re-conversion, and increased to 226 ± 145 μmol /L at 12 months post-reconversion (p = 0.2). Two of them showed a significant increase in mean creatinine level at 12 months post-reconversion (from 179 ± 61 to 377 ± 16 μmol /L, p = 0.04), and one of these 2 patients required hemodialysis at month 9 post-reconversion.

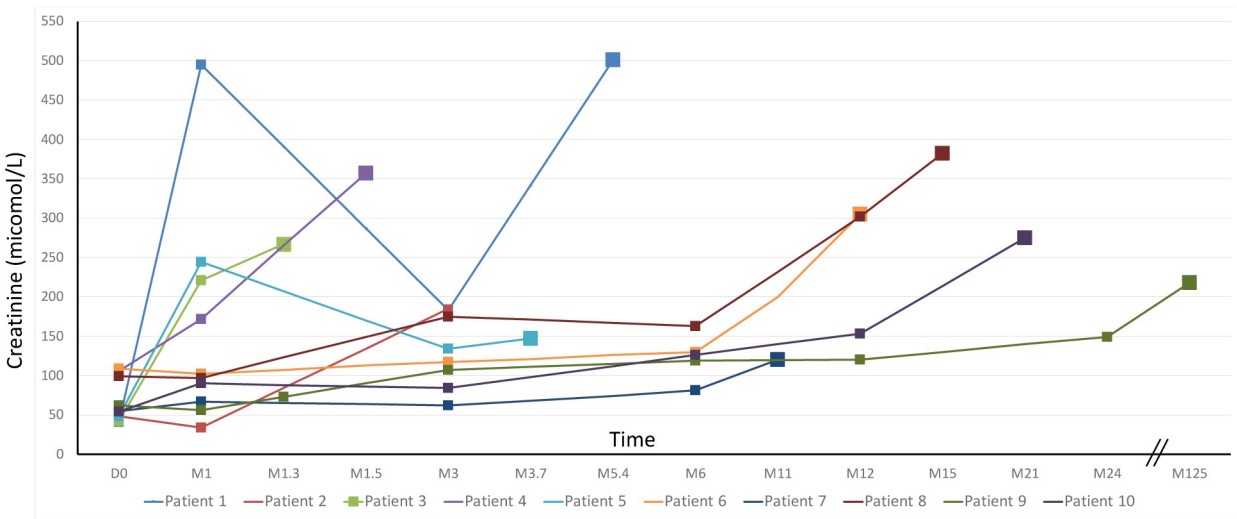

**Fig 1. Post-LTx outcome of individual blood creatinine values in the 10 patients before starting belatacept, from day 0 (D0) of transplantation to the day of conversion to belatacept.**

### ACR episodes after belatacept initiation

ACR episodes occurred in 4/10 patients after belatacept initiation, exclusively in patients with CNI-free belatacept-based IS (Table 2). The TBBx for these 4 patients showed no ACR (A0B0) before starting belatacept. One of the 4 patients experienced ACR grade A1 at months 1 and 3 after starting belatacept, with subsequent normal TBBx results at month 6 (patient 2), and 3 experienced recurrent and/or severe ACR within the first 7 months after belatacept initiation (ACR grade A2 to A4 episodes, patients 3, 6, and 7). These ACR episodes were clinically asymptomatic in 3 cases (patients 2, 3, and 6), whereas patient 7 experienced fulminant ACR grade A4 with acute respiratory distress syndrome (ARDS) at month 1 after belatacept initiation with subsequent related death, which was previously reported [9]. Among the 3 asymptomatic patients with ACR, only 1 patient (patient 3) exhibited new-onset radiology-evidenced involvement, as increasing insidious alveolar opacities within a 2-month period. The other 3 patients with CNI-sparing belatacept IS showed stable function of lung function and no ACR (Table 2). ACR was the cause for the re-switch of belatacept to CNI in 3 patients (patients 3, 6, and 7, Table 3).

### CLAD onset and outcome during belatacept exposure

CLAD onset during belatacept exposure [21] and at last follow-up is detailed in Table 3 for the 9 patients with available follow-up (excluding patient 7 with ARDS-related death from ACR). Two of these patients experienced CLAD onset during belatacept exposure, with mixed-pattern CLAD grade 1 requiring continuation of $O_2$ at exertion already present at belatacept initiation (patient 1) or with a bronchiolitis obliterans syndrome (BOS)1 pattern (patient 4). Another patient with BOS1 at belatacept initiation remained with stable lung function (patient 9). The outcome of individual forced expiratory volume in 1 sec values for patients after starting belatacept is shown in Fig 2B. At the last follow-up after belatacept exposure, death occurred in 2/10 patients with previous stable graft function (1 from fulminant ACR, and 1 sudden death at home with an unknown cause [Table 3]). At the last follow-up post-LTx (median time 27 months [15–52]), 2/10 patients had died, and 4/9 patients had CLAD.

A. Outcome of individual serum creatinine clearance values in the 10 patients after starting belatacept, from day 0 of belatacept to last follow-up under belatacept.

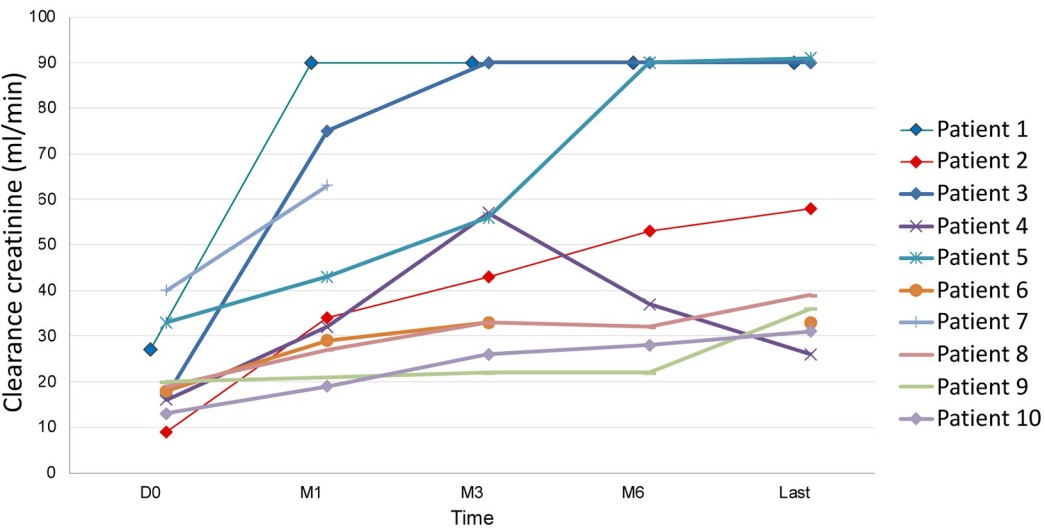

B. Outcome of individual forced expiratory in 1 second (FEV1) values for patients after starting belatacept from day 0 to last follow-up.

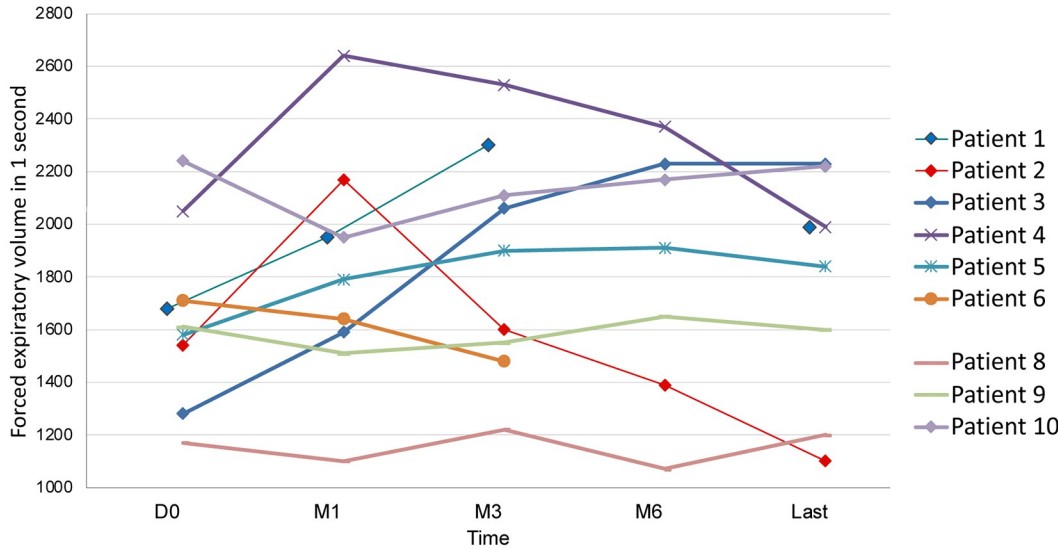

**Fig 2.** A: Outcome of individual serum creatinine clearance values in the 10 patients after starting belatacept, from day 0 of belatacept to last follow-up under belatacept; B: Outcome of individual forced expiratory in 1 second (FEV1) values for patients after starting belatacept from day 0 to last follow-up.

## Viral and opportunistic infections

Viral infection occurred in 3/10 patients under belatacept treatment (Table 3), including 2 cases of disseminated varicella-zoster virus (VZV) infection. Belatacept was discontinued in 1

**Table 2. Acute rejection episodes before and after starting belatacept.**

| Pt n° | Clinical status at belatacept initiation | DSA outcome From day 0 of LTx until date starting belatacept | ACR before starting belatacept | Planned TBB 1 mo after belatacept | Planned TBB 3 mo After belatacept | Planned TBB 6 mo After belatacept | TBB > 6 mo post-belatacept |
|---|---|---|---|---|---|---|---|
| 1 | Post-ICU O2 at exertion | M3: DQ6, MFI 4501. PLEX + IVIg x3 | A0B0 (M1) Definite AMR (M3) | A0B0 | - | A0B0 BOOP | - |
| | | day1 B ELA: DQ6, MFI 3643 | | | | | |
| 2 | Stable | Day 0: DQ7, MFI 4021. day15: DQ7 MFI 13320 | A3Bx (D19) | A1Bx | A1Bx | A0B0 | |
| | | PLEX + IVIg (x4) | | | | | |
| | | day1 BELA: DQ7, MFI 4021 | | | | | |
| 3 | Stable | Day 0: DR52 MFI 2701 PLEX + IVIg (x4) | No ACR | NC | A2B0 | A0BO | A3Bx M7 post-switch) |
| | | day1 BELA: DR52, MFI 1418 | | | | | |
| 4 | Stable | Day 0: no DSA | Clinical AR (M1) | ND | ND | ND | ND |
| | | Day 1 BELA: no DSA | | | | | |
| 5 | Stable | Day 0: no DSA | A0B0 (M3) | ND | ND | ND | ND |
| | | Day 1 BELA: no DSA | A0B0 (M1) | | | | |
| 6 | Stable | Day 0: DQ8 MFI 1109 | A2B0 (M11). | A2B0 | A0B0 | A2B0 | - |
| | | Day 1 BELA: B44 MFI 650 | | | | | |
| 7 | Stable | Day 0: no DSA | No ACR | A4Bx | | | |
| | | Day 1 BELA: no DSA | | DAD | | | |
| | | | | C4d- | | | |
| 8 | Stable | Day 0: no DSA | A1B0 (M9) | - | - | - | - |
| | | Day 1 BELA: no DSA | A1B0 (Y1M1) | | | | |
| 9 | BOS 1 | Day 0: no DSA | No ACR | - | - | - | - |
| | | Day 1 BELA: no DSA | | | | | |
| 10 | BOS 0p | Day 0: no DSA | No ACR | - | - | - | - |
| | | Day 1 BELA: no DSA | | | | | |

DSA: donor-specific antibody. PLEX: Plasma exchange; IVIg: intravenous immunoglobulins; ACR: acute cellular rejection; BOS: bronchiolitis obliterans syndrome; DSA: donor-specific antibody; BELA: belatacept; ACR: acute cellular rejection; A et B: grade A and grade B of ACR; TBBx: transbronchial biopsies; M: month; Y: year; ND: not done; NC: not contributive; Clinical AR: deterioration of lung function regressive after high dose (boluses) of IV methylprednisolone. TBB: transbronchial biopsies; Planned TBB at 1, 3, and 6 months after starting belatacept, when feasible, were performed only in patients 1, 2, 3, 4, 5, 6, and 7 followed at Bichat center. TBB post-belatacept conversion was performed only for clinical indication in patients 8,9, and 10 (Foch center).

of these 2 patients with VZV infection. Pneumocystosis occurred in one patient (patient 10), who was suspected of non-adherence to atovaquone treatment.

## Discussion

In this series of LTx recipients with CNI-related severe renal insufficiency, the use of belatacept was associated with significant improvement in kidney function but also with a high incidence of recurrent/severe ACR, only observed in patients with CNI-free belatacept regimen (4/7 patients). Two of 10 patients also experienced severe viral disease, as reported in other solid-organ Tx cases under belatacept [6, 19].

The significant improvement in kidney function with belatacept is in accordance with the potential partial reversibility of severe renal insufficiency due to CNI toxic effects observed in kidney Tx [22, 23], in heart Tx [24] and in some LTx reports [11, 12]. This improvement was mainly observed with CNI-free belatacept IS [12] but also with CNI-sparing belatacept IS [11] with very low-dose CNI in our patients.

**Table 3. Incidence of infections and chronic allograft rejection (CLAD) after starting belatacept.**

| Pt no | Duration of belatacept exposure | Infection under belatacept | CLAD onset under belatacept | DSA outcome From day 1 belatacept to last detection of DSA under belatacept | Time point and cause of discontinuation of belatacept | IS after belatacept discontinuation | Status at last follow-up Post-LTx |
|---|---|---|---|---|---|---|---|
| 1 | 12M | -Disseminated VZV (M1) -Pneumonia (Staphylococcus) | Yes CLAD1 (mixed) | day 1: DQ6, MFI 3643 M10: no DSA | continued | - | CLAD, Mixed pattern M18 |
| 2 | 5M | no | No (STA) | day 1: DQ7, MFI 4021 M5: no DSA | continued | - | Stable at death M9 (sudden death from unknown cause at home). |
| 3 | 8M | no | No (STA) | day 1: DR52, MFI 1418 M6: DR52, MFI 1795 | M8 Recurrence of high grade ACR episodes (A2, A3) | Ciclosporine: (80–150 ng/ml) MMF 2g/d Pred 5 mg/d | Stable M16 |
| 4 | 21M | no | Yes BOS1 | day 1: no DSA M16: no DSA | M21. indication for CNI due to worsening FSG + BOS1 onset + resistant CMV | FK:4–7 ng/ml MMF 1g/d Pred 5 mg/d | BOS3 M21 |
| 5 | 25M | PCR HSV1 + in BALF (M2) asymptomatic | No (STA) | day 1: no DSA M6, M12: no DSA | M25 change in LTx IS policy | FK:6–9 ng/ml MMF 1g/d Pred 5 mg/d | Stable M49 |
| 6 | 4M | no | No (STA) | day 1: B44 MFI 650 M3: no DSA | M4 Recurrent ACR (A2, A2) | FK:4–7 ng/ml MMF 2g/d Pred 5 mg/d | BOS 1 M64 |
| 7 | 1.5M | no | NE | day 1: no DSA M1: no DSA | Day 45 Fatal severe ACR | FK:4–7 ng/ml MMF 2g/d Pred 5 mg/d | Death M13 (ACR) |
| 8 | 23M | no | No (STA) | day 0: no DSA | continued | - | Stable M39 |
| 9 | 12M | no | Stable BOS1 | day 0: DSA | continued | - | BOS1 Y11M5: 137M |
| 10 | 12M | -pneumocystosis (M4 belatacept) -Disseminated VZV (M12 belatacept) | No (STA) | day 0: no DSA | M12. Recurrent opportunistic infections | FK:6–9 ng/ml MMF 1g/d Pred 5 mg/d | Stable Y2M10: 34M |

VZV: varicella zoster virus; ACR: acute cellular rejection episode. M: month; Y, year; BALF: bronchoalveolage fluid; VZV: varicella zoster virus; HSV, herpes simplex virus; CLAD: chronic lung allograft dysfunction; BOS: bronchiolitis obliterans syndrome; CMV: cytomegalovirus; DSA: donor-specific antibody; MFI: mean fluorescent intensity; CNI; calcineurin inhibitor; FSG: focal segmental glomerulosclerosis; ACR: acute cellular rejection episode; DSAs: donor-specific antibodies; MMF: mycophenolate mofetil, FK: tacrolimus; values in ng/ml following ciclosporine and FK correspond to blood target of C0 for ciclosporine and FK; d: day.

The 2 cases of severe viral infections with disseminated VZV disease each occurred in one of the 2 centers, with or without previous induction IS therapy administered at the date of LTx. The severity of VZV infection led to discontinuing belatacept in 1 patient. Because VZV infection may present in a threatening form after solid-organ transplantation [25], this suggests that a systematic prophylaxis against VZV infection could be used in case of a positive serologic status in post-LTx under belatacept.

The occurrence of ACR episodes and/or CLAD development remain the main potential risk after belatacept conversion. The frequency of high grade ACR we observed in this series seems of particular concern. Of note, these ACR episodes occurred exclusively among the 7 patients with CNI-free belatacept IS (n = 4/7), with an unusual clinical presentation: indolent progressive ACR episodes or, in contrast, a fulminant ARDS-associated ACR episode [9]. The few previous LTx reports of CNI-free belatacept IS conversion also described an unusually high rate of ACR episodes: 5/9 evaluable patients in the series reported by Iasella et al. [12] or in 3 other reported clinical cases, including ACR with functional decline [10], highly suspected ACR [8], or severe ACR with high-grade lymphocytic bronchiolitis [14]. In contrast, the 3 patients with CNI-sparing belatacept IS in our series experienced no ACR episode, but these

patients initiated belatacept after the first year post-LTx, a period with lower risk of ACR as compared with the first year post-LTx.

CLAD occurred in 2/9 patients under belatacept with available follow-up after starting the conversion from CNI to belatecept, only in those under CNI-free belatacept IS (Table 3). Hence, we can draw no firm conclusion on risk of CLAD from these preliminary experiences with CNI-sparing belatacept IS (our series, [12]). In parallel, in 4/10 patients with DSAs detected at belatacept initiation, 3 showed DSA clearance. This observation also does not allow for any conclusion because of the low number of patients but agrees with findings in kidney Tx showing fewer DSAs in recipients under belatacept [7].

Belatacept was discontinued in 6 patients who were re-switched to a standard CNI-based IS. The causes of discontinuation were mainly recurrent/severe ACR in 3 patients with CNI-free IS but also severe viral infection in 1 patient, an indication for CNI treatment in 1 patient (because of worsening focal segmental glomerulosclerosis),or change in center policy in 1 patient. Among 5 patients reconverted to CNI and with available follow-up, two experienced again a significant decline in renal function, which suggests also the potential benefit of belatacept for kidney function in case of CNI toxic effects after LTx.

The occurrence of one case of rapid-onset ARDS-related ACR in the CNI-free belacept IS group led to modifying our protocol for using belatacept, with systematic adjunction of very-low dose CNI in the next 3 patients. Our hypothesis was that the very-low dose CNI would be able to control the T-cell subsets potentially involved in belatacept-resistant rejection and to reverse or at least stabilize the CNI toxic effects. The 3 other patients who received CNI-sparing belatacept IS experienced no ACR episode or CLAD onset, but these 3 patients initiated belatacept only after the first year post-LTx, in a period with lower risk of ACR. Another series of 8 patients with CNI-sparing belatacept IS with ongoing CNI administered at a very low dose [11] also showed subsequent stable lung function and low incidence of ACR episodes (only 1 ACR episode among 8 patients), which suggests a possible benefit of such CNI-sparing belatacept IS regimen. Besides these single-center reports of belatacept conversion, belatacept was also recently investigated in a pilot randomized controlled trial of LTx as *de novo* belatacept-based IS versus tacrolimus-based IS. The belatacept group exhibited a significant increase in early deaths, with premature termination of the trial; however, both arms exhibited the same incidence of ACR, *de novo* DSAs, or CLAD occurrence [26], which led to lack of definitive conclusions for increased risk of rejection due to belatacept. Notably, causes of deaths in the belatacept arm included non-immunologic deaths early after LTx, so the definitive imputability of belatacept remained uncertain. Experience with CNI-free belatacept IS regimen has also been reported after heart Tx, showing, as in our series, more frequent and more severe rejection episodes after conversion to belatacept [24].

Taken together, our and previous studies suggest a high incidence of alloimmune-mediated injury associated with CNI-free belatacept IS after LTx during the first year [8, 10, 12, 14] that was possibly linked to belatacept-resistant ACR mechanisms already reported in kidney Tx [21, 27]. The use of belatacept-based IS has been found a viable option in large phase-3 kidney Tx studies, with similar graft survival as compared with CsA-based IS [7, 28]. Nevertheless, other studies comparing *de novo* belatacept-based IS with tacrolimus-based IS in kidney-Tx recipients at high immunological risk showed a significant increase in ACR rate and grades in the belatacept arm [21, 27]. Of note, the addition of a transient course of tacrolimus with belatacept IS reduced rejection rates to acceptable levels, similar to those of the tacrolimus group [27]. The hypothesis for explaining these belatacept-resistant ACR cases in kidney Tx suggests increased subsets of CD28[neg] T cells able to escape belatacept's mechanism of action due to the lack of CD28 target, with the potential use of the pre-Tx percentage of CD28[pos] memory T cells to predict expansion of CD28[neg] T cells associated with subsequent belatacept-resistant

ACR after Tx [29–31]. In the absence of clinical validation of such predictors, our results suggest to further explore CNI-sparing belatacept IS rather than CNI-free IS, after a case-by-case careful evaluation of the risk/benefit ratio. Interestingly, a small series in kidney Tx with a high immunological risk also showed satisfactory outcome with a very similar protocol that we used (CNI-sparing belatacept IS in association of same low dose Tacrolimus [2–3 ng/mL]) [32]. It should be emphasized that our small series of heterogeneous LTx recipients does not allow firm conclusion, and only suggests a possible benefit of CNI-sparing belatacept in low-immunological risk patients after 1 year post-Tx.

In summary, belatacept has been considered an attractive IS candidate drug for use in case of severe and/or life-threatening CNI toxic effects after solid-organ Tx.

Nevertheless, this series of lung-Tx recipients showed a high incidence of recurrent/severe ACR episodes under belatacept that were a particular concern in patients starting a CNI-free belatacept-based IS early after LTx. The high immunological risk we observed after LTx suggests the need for a careful clinical assessment of the risk/benefit ratio between the probability of improving CNI toxic effects versus the risk of belatacept-resistant rejection before starting belatacept in this population. Other studies are needed to better determine the potential indications of the use of belatacept after LTx, possibly with lower immunological-risk IS regimens, such as CNI-sparing belatacept IS after the first year post-LTx.

## Supporting information

**S1 Data.**
(DOC)

## Author Contributions

**Conceptualization:** Olivier Brugière, Quentin Raimbourg, Hervé Mal, Vincent Bunel.

**Formal analysis:** Vincent Bunel.

**Investigation:** Marie-Noelle Peraldi, Sylvie Colin de Verdière, Laurence Beaumont, Abdulmonem Hamid, Mathilde Zrounba, Antoine Roux, Clément Picard, François Parquin, Matthieu Glorion, Julie Oniszczuk, Alexandre Hertig.

**Methodology:** Olivier Brugière, Alexandre Vallée, Hervé Mal, Vincent Bunel.

**Writing – original draft:** Olivier Brugière.

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
