## [Decision Letter · Decision Letter 0]

20 Oct 2022

PONE-D-22-26417Conversion to belatacept after lung transplantation: report of 10 casesPLOS ONE

Dear Dr. Brugière,

Thank you for submitting your manuscript to PLOS ONE. After careful consideration, we feel that it has merit but does not fully meet PLOS ONE’s publication criteria as it currently stands. Therefore, we invite you to submit a revised version of the manuscript that addresses the points raised during the review process.

We look forward to receiving your revised manuscript.

Kind regards,

Niels Olsen Saraiva Câmara, M.D, PhD

Academic Editor

PLOS ONE

Journal Requirements:

Additional Editor Comments:

Dear Authors,

Your manuscript has been evaluated and we are asking you to revise the text accordingly to peer review. Special attention should be done for final text format.

Reviewers' comments:

Reviewer's Responses to Questions

**Comments to the Author**

1. Is the manuscript technically sound, and do the data support the conclusions?

Reviewer #1: Yes

Reviewer #2: Yes

2. Has the statistical analysis been performed appropriately and rigorously? 

Reviewer #1: Yes

Reviewer #2: N/A

3. Have the authors made all data underlying the findings in their manuscript fully available?

Reviewer #1: Yes

Reviewer #2: Yes

4. Is the manuscript presented in an intelligible fashion and written in standard English?

Reviewer #1: Yes

Reviewer #2: Yes

5. Review Comments to the Author

Reviewer #1: The present study is a retrospective case series of 10 lung transplant recipients treated with belatacept almost exclusively for renal failure related to calcineurin inhibitor toxicity. This report is important to refine the possibility of using belatacept in these patients, perhaps with a CNI-sparing rather than a CNI free strategy, and after the first post-transplantation year. Indeed, it clearly confirms a sharp improvement in renal function, but also suggests a high risk of opportunistic infections and probable rejections.

Major comments

The improvement in renal function is impressive and well described. However, we lack information following conversion from belatacept to CNI (n=6). Did renal function remain stable or decline again?

The proportion of rejections is high following first-year conversion from CNI to belatacept. We need to know:

- Dose of MMF and AUC if available after conversion

- Is tacrolimus C0 target 2-3 ng/ml or 1-3 ng/ml? (two different information in the text)

- Induction (Basiliximab, ATG or none) should be added in Table 1

- History sensitization (DSA?) before conversion could be inserted in Table 2

Several opportunistic infection are reported. This is a crucial and worrying point already described in other types of solid organ transplantations. Did they use post-conversion prophylactic treatment (for CMV and pneumocystosis). In addition, can authors provide information on D/R CMV serology as well as VZV serology? I find that discussion is essentially centered on rejection while infectious risk is just as important.

Minor comments

- I alert authors that some surnames and family names are inverted

- The meaning of abbreviation ACR should be provided at the first appearance in the text (present only in the abstract)

Reviewer #2: The authors present a unique case series in lung transplant recipients with CNI associated renal toxicity converted to Belatacept immunosuppression. Conversion had the intended effect on renal function but also introduced higher than desired rates of acute rejection. Overall, the manuscript mostly meets PLOS ONE criteria and the following should enhance the manuscript further.

1. The diction and grammar, use of language is not as clear as it should be for final publication. understandably English is presumably not the first language of the French authors, but improvements in this will enhance the readability and communication of data.

2. Better disclosure of baseline renal function of subjects reported on would be helpful. It was stated they had normal eGFR (except 1) pre-transplant, but outlining this at least in the table and maybe even in graphical form would be very helpful. Not only pre-tx but also over the course post-transplant before conversion. I would also suggest the eGFR graph be primary figure and not supplementary, as it is most of the story here.

3. Improvements in eGFR were quite dramatic. Given that they were all relatively early post-tx conversions, it is possible other transplant or or early post-tx factors were contributing to the low eGFRs in addition to CNI. this is why demonstrating the eGFR course during the post-tx, pre-conversion period might be informative. As well as addressing this possibility in the data presentation or discussion since controls are not part of the nature of case series such as this one.

4. What happened to the eGFRs once patients were put back on CNI due to ACR?

5. Disclosing the induction IS may also be informative re: rejection risk for each patient. Were the non-rejectors thymoglobulin recipients and rejectors Belatacept? were all recipients on triple IS (? prednisone)

6. Emphasis on the low dose CNI approach with Belatacept as possible future approach should be stronger. Early post-tx risk of rejection with Belatacept generally understood across organs, and best highlighted by the Emory experience and BEST study in post trial experiences. mitigating with low dose cni/bela approach very rationale and slowly achieves goal of longer term preservation of renal function.

7. Reference and discussion of experience with Bela in the other cardiothoracic organ (heart) may also complete discussion on the subject for this case series. several publications exist on conversion in this setting as well, but with less rejection as I recall, but this should be verified.

8. the authors should very clearly state that this is a small case series composed of heterogeneous patient population that suggests certain points, but conclusions cannot be made.

6. PLOS authors have the option to publish the peer review history of their article (what does this mean?). If published, this will include your full peer review and any attached files.

Reviewer #1: No

Reviewer #2: No

---

## [Author Response · Author response to Decision Letter 0]

21 Dec 2022

RESPONSES REVIEWERS COMMENTS 

PONE-D-22-26417

Conversion to belatacept after lung transplantation: report of 10 cases

PLOS ONE

Reviewer #1: The present study is a retrospective case series of 10 lung transplant recipients treated with belatacept almost exclusively for renal failure related to calcineurin inhibitor toxicity. This report is important to refine the possibility of using belatacept in these patients, perhaps with a CNI-sparing rather than a CNI free strategy, and after the first post-transplantation year. Indeed, it clearly confirms a sharp improvement in renal function, but also suggests a high risk of opportunistic infections and probable rejections.

Major comments

The improvement in renal function is impressive and well described. However, we lack information following conversion from belatacept to CNI (n=6). Did renal function remain stable or decline again?

We agree this is interesting to know if what was the renal function outcome after re-conversion from belatacept to CNI. Among these 6 patients reconverted from belalacept to CNI, outcome of renal function was available at 12 months post-re-conversion in 5 of them, because one of them died rapidly thereafter (M1.5 post-reconversion, patient 7). In the 5 patients with available follow-up, creatinine value was 154 (SD 69) micromole/L at the date of re-conversion, and increased to 226 (145) micromole/L at 12 months post-reconversion (p=0.2). Two of these 5 patients had significant increase of mean creatinine value from the date of re-conversion belatacept-tacrolimus to 12 months post-reconversion (from 179 (61) micromole/L to 377 micromol/L, p=0.04), and one of these 2 patients required hemodialysis at month 9 post-reconversion. Blood C0 target of tacrolimus after reconversion was 4-7 ng/ml (patient 4 and 6), 6-9 ng/ml (patient 5 and 10), and blood C0 target of ciclosporine was 80-150 ng/ml (patient 3). Hence, although the very few number does not allow for firm conclusion, the renal outcome the first year after re-conversion to CNI showed a severe decline of kidney function in 2 patients, and hemodialysis was required in 1 of them, suggesting again the transient benefit of belatacept exposure on kidney function in case of CNI toxicity after LTx.

As suggested, we added these data in the manuscript, section renal outcome after belatacept page 7, line 12: 

“Belatacept was discontinuated in 6 patients who were re-switched to a standard CNI-based IS. Causes of discontinuation were: recurrent/severe ACR [n=3], viral infection [n=1], change in center policy [n=1], other [n=1]). In the 5 patients with available follow-up after belatacept discontinuation, creatinine value was 154 (SD 69) micromole/L at the date of re-conversion, and increased to 226 (145) micromole/L at 12 months post-reconversion (p=0.2). Two of them had significant increase of mean creatinine value at 12 months post-reconversion (from 179 [61] to 377 [16] micromol/L, p=0.04), and one of these 2 patients required hemodialysis at month 9 post-reconversion. “

In Table 3, IS regimen of patients who were re-converted from belatacept to CNI are now included, to detailed the dosage of CNI during this period (See Table 3).

In discussion page 12 line 16:

“A significant decline of renal function was observed in 2 among 6 patients who were reconverted from belatacept to CNI, which again suggests the potential benefit of belatacept on kidney function in case of CNI toxicity following LTx.”

The proportion of rejections is high following first-year conversion from CNI to belatacept. We need to know:

- Dose of MMF and AUC if available after conversion

Dosage of MMF after conversion from CNI to belatacept was as follows: 1000 mgx2/day (patients 2, 3, 6, and 7) or 500mgx2/day (patients 1, 4, and 5). No AUC was performed for these patients./ As suggested, these data have been inserted in the last column of Table 1 (see Table 1).

- Is tacrolimus C0 target 2-3 ng/ml or 1-3 ng/ml? (two different information in the text)

Tacrolimus C0 target was 2-3 ng/ml. This has been corrected.

- Induction (Basiliximab, ATG or none) should be added in Table 1.

Induction therapy was not used for patients from Bichat (n=7). Nevertheless, an early use of basiliximab was used in the patients 1, 3, and 5 in this center after the immediate postoperative period, with an indication “CNI holiday”. This CNI holiday consisted in transient CNI reduction dosage associated with the use of basilixilab, at M3 for patient 1, M1 for patient 3, and M3 for patient 5. In case of CNI holiday, basiliximab 20 mg/d at day 0 and day 4 was administered, associated with reduction of CNI dosage (Tacrolimus C0: 2-3 ng/ml during 10 days). Because basiliximab was used in these patients, not as induction but as CNI holiday, it has been mentioned and inserted in Table 1.

An induction therapy was used for patients from Foch hospital in 2 patients (Thymoglobulins), and these data have been inserted in Table 1. 

- History sensitization (DSA?) before conversion could be inserted in Table 2

As suggested, history of sensitization before the date of starting belatacept in now shown in Table 2, associated to desensitization treatment administered to these sensitized patients. For example, Patient 1 had progressive increase of de novo DSA (anti-DQ6 with MFI of 4501 at M3), treated with PLEX and monthly IVIg (x3) before the date belatacept was started (See Table 2). 

This has been added in Table 2. 

Several opportunistic infection are reported. This is a crucial and worrying point already described in other types of solid organ transplantations. Did they use post-conversion prophylactic treatment (for CMV and pneumocystosis). In addition, can authors provide information on D/R CMV serology as well as VZV serology? I find that discussion is essentially centered on rejection while infectious risk is just as important.

-Prophylactic treatment against pneumocystosis was systematically used in all patients (in both centers) from day 0, with Trimetoprim/Sulfamethoxasone (800mg): 800 mg 3 times a week, or atovaquone 750 mg x 2/day in case of side effects of Trimetoprim/Sulfamethoxasone, lifelong from day 0 of LTx. One patient had pneumocystosis infection at M4 after starting belatacept treatment. The patient was under atovaquone at this date, and it was suspected that the adherence to atovaquone was not optimal at this post-LTx date in this patient. This has been specified in the manuscript.

Manuscript, page 11 line 5: “Pneumocystosis occurred in one patient (patient 10), who was suspected of non-adherence to atovaquone treatment.”.

And in Supplemental data, page 2 line 5: “Prophylactic treatment against pneumocystosis was systematically used in all patients (in both centers) from day 0, with Trimetoprim/Sulfamethoxasone (800mg): 800 mg 3 times a week, or atovaquone 750 mg x 2/day in case of side effects of Trimetoprim, lifelong from day 0 of LTx”.

-Prophylactic treatment of CMV infection was not altered by the conversion CNI-Belatacept, whatever the date of starting belatacept. At Bichat hospital, it included Rovalcyte: 900 mg/day from day 0 post LTx to 6 months in absence of CMV mismatch or to 12 months post-LTx in case of CMV mismatch. At Foch hospital, same prophylactic treatment was used, except in patients with CMV D+/R+ or D-/R+, for who Zelitrex 1000 mg x 4/day was used during the first 6 months. In both centers, in case of negative CMV status (D-/R-), a prophylactic treatment with Zelitrex 1000 mg x3/day was given within the first 3 months. 

As suggested, D/R CMV serology and VZV serology are now detailed in the manuscript (Table 1). In supplemental data, we have now specified prophylaxis for CMV and VZV infections, page 2 line 14: “Prophylactic treatment of CMV infection at Bichat hospital included Rovalcyte: 900 mg/day from day 0 post LTx to 6 months in absence of CMV mismatch, or to 12 months post-LTx in case of CMV mismatch. At Foch hospital, same prophylactic treatment was used, except in patients with CMV D+/R+ or D-/R+ status, for who Zelitrex 2000 mg x 4/day was used during the first 6 months. In both centers, in case of negative CMV status (D-/R-), a prophylactic treatment with Zelitrex 1000 mg x3/day was given within the first 3 months.”

We agree with reviewer that occurrence of severe opportunistic infections under belatacept is a crucial point also, as showed in 2 patients with disseminated VZV infection (patient 1 and 10 in this series, which led to discontinuation of belatacept in one of them (patient 10). In these 2 patients, VZV serology of recipient was positive before Tx (inserted in Table 1). These 2 patients were without viral prolonged specific prophylactic treatment of VZV infection at the date of starting belatacept (day 162 and Y1M9, respectively), but patient 1 had only the end of Rovalcyte prophylaxis during the first 20 days under belatacept. Hence, the severity of these disseminated VZV disease under belatacept, as previously described in SOT recipients, raises this issue of systematic prophylaxis against VZV infection in case of a positive serologic status in post-LTx. For CMV disease, systematic surveillance of CMV PCR may allow to early detect CMV replication without systematic prophylaxis and to avoid severe form of infection with adapted rovalcyte treatment.

As suggested, we have now highlighted this issue in the manuscript:

-Table 3: D/R CMV serology and VZV serology are now detailed in the manuscript (Table 1).

-In the discussion, page 11 line 22: “Hence, because VZV infection may present as a life threatening disseminated infection after solid-organ transplantation (1), this suggests to use a systematic prophylaxis against VZV infection in case of a positive serologic status in post-LTx.”

Minor comments

- I alert authors that some surnames and family names are inverted

We thank reviewer 1 and this has been corrected.

- The meaning of abbreviation ACR should be provided at the first appearance in the text (present only in the abstract).

This has been corrected.

Reviewer #2: The authors present a unique case series in lung transplant recipients with CNI associated renal toxicity converted to Belatacept immunosuppression. Conversion had the intended effect on renal function but also introduced higher than desired rates of acute rejection. Overall, the manuscript mostly meets PLOS ONE criteria and the following should enhance the manuscript further.

1. The diction and grammar, use of language is not as clear as it should be for final publication. understandably English is presumably not the first language of the French authors, but improvements in this will enhance the readability and communication of data.

As suggested, the whole manuscript has been reviewed by an English native.

2. Better disclosure of baseline renal function of subjects reported on would be helpful. It was stated they had normal eGFR (except 1) pre-transplant, but outlining this at least in the table and maybe even in graphical form would be very helpful. Not only pre-tx but also over the course post-transplant before conversion. I would also suggest the eGFR graph be primary figure and not supplementary, as it is most of the story here.

As suggested, outcome of renal function starting from day 0 of LTx to the day starting belatacept for all patients has now been added with a new Figure 1, as a primary figure (see Figure 1: Outcome of individual creatinine values for all patients before starting belatacept from day 0 of lung transplantation to the day starting belatacept).

In the manuscript, page 7 line 2: “Post-LTx outcome of renal function before conversion to belatacept for all patients is shown in Figure 1, as individual creatinine values of the 10 patients from day 0 of LTx to the day of conversion to belatacept. Mean creatinine value increased from 66 (27) micromole/L at day 0 of LTx to 275 (116) micromole/L at the date of conversion to belatcept (p=0.0002).”

In the same way, and as suggested, the previous supplemental figure S1, showing the outcome of individual creatinine clearance values for patients after starting belatacept from day 0 of belatacept to last follow-up, is now shown as a primary figure (Figure 2A), with an addition of the following sentence page 7, line 17 in the manuscript (section results): “Outcome of individual creatinine clearance values for all patients under belatacept is shown in Figure 2A. Among all patients, mean creatinine value decreased from 274 (116) �mol at the date of conversion to belatcept to 126 (65) �mol at the last day of follow-up under belatacept (p=0.0009).”

3. Improvements in eGFR were quite dramatic. Given that they were all relatively early post-tx conversions, it is possible other transplant or or early post-tx factors were contributing to the low eGFRs in addition to CNI. this is why demonstrating the eGFR course during the post-tx, pre-conversion period might be informative. As well as addressing this possibility in the data presentation or discussion since controls are not part of the nature of case series such as this one.

As suggested in response to point n°2, the pre-conversion period of renal function outcome is now shown in data, and included in a new Figure 1 as a primary Figure (see response to point 2).

4. What happened to the eGFRs once patients were put back on CNI due to ACR?

We agree with reviewer that this is an important point. Among these 6 patients reconverted from belalacept to CNI, outcome of renal function was available at 12 months post-re-conversion in 5 of them, because one of them died rapidly thereafter (M1.5 post-reconversion, patient 7). In the 5 patients with available follow-up, creatinine value was 154 (SD 69) micromole/L at the date of re-conversion, and increased to 226 (145) micromole/L at 12 months post-reconversion (p=0.2). Two of these 5 patients had significant increase of mean creatinine value from the date of re-conversion belatacept-tacrolimus to 12 months post-reconversion (from 179 (61) micromole/L to 377 micromol/L, p=0.04), and one of these 2 patients required hemodialysis at month 9 post-reconversion. Blood C0 target of tacrolimus after reconversion was 4-7 ng/ml (patient 4 and 6), 6-9 ng/ml (patient 5 and 10), and blood C0 target of ciclosporine was 80-150 ng/ml (patient 3). Hence, although the very few number does not allow for firm conclusion, the renal outcome the first year after re-conversion to CNI showed a severe decline of kidney function in 2 patients, and hemodialysis was required in 1 of them, suggesting again the transient benefit of belatacept exposure on kidney function in case of CNI toxicity after LTx.

As suggested, we added these data in the manuscript, section “renal outcome” page 7, line 12: 

“Belatacept was discontinuated in 6 patients who were re-switched to a standard CNI-based IS (causes of discontinuation were: recurrent/severe ACR [n=3], viral infection [n=1], change in center policy [n=1], other [n=1]). In the 5 patients with available follow-up after belatacept discontinuation, creatinine value was 154 (69) micromole/L at the date of re-conversion, and increased to 226 (145) micromole/L at 12 months post-reconversion (p=0.2). Two of them had significant increase of mean creatinine value at 12 months post-reconversion (from 179 [61] to 377 [16] micromol/L, p=0.04), and one of these 2 patients required hemodialysis at month 9 post-reconversion. 

In Table 3, IS regimen of patients who were re-converted from belatacept to CNI are now included, to detail the dosage of CNI during this period (See Table 3).

In discussion page 12 line 16. :

“A significant decline of renal function was observed in 2 among 5 patients who were reconverted from belatacept to CNI, which again suggests the potential benefit of belatacept on kidney function in case of CNI toxicity following LTx.”

5. Disclosing the induction IS may also be informative re: rejection risk for each patient. Were the non-rejectors thymoglobulin recipients and rejectors Belatacept? were all recipients on triple IS (? prednisone).

ACR episodes occurred in 4/10 patients after belatacept initiation, exclusively in patients with CNI-free belatacept-based IS (Table 2). These 4 patients did not receive induction therapy by thymoglobulin, and this has been added, in Table 1 and in the manuscript. Nevertheless, although induction therapy was not used for patients from Bichat (n=7), an early use of basiliximab was used during the first year in patients 1, 3, and 5 of this center. This indication of “CNI holiday” consisted in transient CNI reduction dosage associated with the use of basilixilab, at M3 for patient 1, M1 for patient 3, and M3 for patient 5. In case of CNI calcineurin holiday, basiliximab 20 mg/d at day 0 and day 4 was administered, associated with reduction of CNI dosage (Tacrolimus C0: 2-3 ng/ml during 10 days). As suggested, all these data have been inserted in Table 1.

An induction therapy was used for patients from Foch hospital (see Table 1 and supplemental data). 

All patients were under triple IS, including belatacept and prednisone, and at least one other IS therapy (Table 1). To be clearer, prednisone and other associated IS therapy have been specified in Table 1.

6. Emphasis on the low dose CNI approach with Belatacept as possible future approach should be stronger. Early post-tx risk of rejection with Belatacept generally understood across organs, and best highlighted by the Emory experience and BEST study in post trial experiences. mitigating with low dose cni/bela approach very rationale and slowly achieves goal of longer term preservation of renal function.

We agree that it is a crucial point. Interestingly, an experience in kidney Tx was reported with the adoption of belatacept-based IS in association of same low dose Tacrolimus (2-3 ng/mL) that we used, showing a satisfactory outcome under this IS regimen in a specific population of KTs with a high immunological risk ( PLOS One 2020, Gallo, E et al. Oct 15 : Prevention of acute rejection after rescue with Belatacept by association of low-dose Tacrolimus maintenance in medically complex kidney transplant recipients with early or late graft dysfunction). This reference has been added and discussed, page 15, line 4: “Interestingly, a small series in kidney Tx with a high immunological risk showed satisfactory outcome with a very similar protocol that we used (CNI-sparing belatacept IS in association of same low dose Tacrolimus [2-3 ng/mL])(2).”

Hence, we believe from our observations that there is a rational for a future approach with very low CNI/belatacept association, and this has been further discussed in the manuscript, (page, line), keeping in mind the point 8 (see below) that this is a small series with heterogeneous patient population which included only 3 patients with CNI-reduction associated with belatacept.

In the discussion, page 14 line 21: “In the absence of clinical validation of such predictors, our results suggest to further explore CNI-sparing belatacept IS rather than CNI-free IS, after a case-by-case careful evaluation of the risk/benefit ratio.”

In the discussion, page 15 line 5, the following sentence: “studies are needed to better determine the potential indications of its use following LTx, possibly with lower immunological risk IS regimens, such as CNI-sparing belatacept IS after the first year post-LTx.”

7. Reference and discussion of experience with Bela in the other cardiothoracic organ (heart) may also complete discussion on the subject for this case series. several publications exist on conversion in this setting as well, but with less rejection as I recall, but this should be verified.

We have now added experience of belatacept IS regiment reported in heart Tx in a series of 40 patients (Launay, M. et al, AJT 2019: Belatacept‐based immunosuppression: A calcineurin inhibitor‐sparing regimen in heart transplant recipients). This study showed rather similar findings as compared to our series in LTx: 1/ The main reason for switching to belatacept was to preserve renal function, resulting in discontinuation of CNI and changes in IS therapy in 76% of cases. Following administration of belatacept, a very significant improvement of renal function was observed (+59%, p=0.0002) in a similar range with our study. 2/More frequent and more severe rejection episodes were observed after conversion to belatacept. One patient died of severe organ rejection while on IS regimen free of CNI, and including belatacept, everolimus, MMF, and steroids. 3/ A rather high number of discontinuation of belatacept was observed, including treatment failure (16/40 during the period of the study).

Hence, we have added these points and this reference in the manuscript:

Page 12 line 1: “The significant improvement in kidney function with belatacept is in accordance with the potential partial reversibility of severe renal insufficiency due to CNI toxic effects observed …, in heart Tx (3) and in some LTx reports (4, 5).

Page 14, line 11: “Experience with CNI-free belatacept IS regimen has also been reported after heart Tx, showing, as in our series, more frequent and more severe rejection episodes after conversion to belatacept (3).”

8. the authors should very clearly state that this is a small case series composed of heterogeneous patient population that suggests certain points, but conclusions cannot be made.

This has been stated in the manuscript page 15, line 10: “It should be emphasized that our small series of heterogeneous LTx recipients does not allow firm conclusion, and only suggests a possible benefit of CNI-sparing belatacept in low-immunological risk patients after 1 year post-Tx.”

1. Helou E, Grant M, Landry M, Wu X, Morrow JS, Malinis MF. Fatal case of cutaneous-sparing orolaryngeal zoster in a renal transplant recipient. Transpl Infect Dis 2017; 19.

2. Gallo E, Abbasciano I, Mingozzi S, Lavacca A, Presta R, Bruno S, Deambrosis I, Barreca A, Romagnoli R, Mella A, Fop F, Biancone L. Prevention of acute rejection after rescue with Belatacept by association of low-dose Tacrolimus maintenance in medically complex kidney transplant recipients with early or late graft dysfunction. PLoS One 2020; 15: e0240335.

3. Launay M, Guitard J, Dorent R, Prevot Y, Prion F, Beaumont L, Kably B, Lecuyer L, Billaud EM, Guillemain R. Belatacept-based immunosuppression: A calcineurin inhibitor-sparing regimen in heart transplant recipients. Am J Transplant 2020; 20: 553-563.

4. Timofte I, Terrin M, Barr E, Sanchez P, Kim J, Reed R, Britt E, Ravichandran B, Rajagopal K, Griffith B, Pham S, Pierson RN, 3rd, Iacono A. Belatacept for renal rescue in lung transplant patients. Transpl Int 2016; 29: 453-463.

5. Iasella CJ, Winstead RJ, Moore CA, Johnson BA, Feinberg AT, Morrell MR, Hayanga JWA, Lendermon EA, Zeevi A, McDyer JF, Ensor CR. Maintenance Belatacept-Based Immunosuppression in Lung Transplantation Recipients Who Failed Calcineurin Inhibitors. Transplantation 2018; 102: 171-177.

---

## [Decision Letter · Decision Letter 1]

24 Jan 2023

Conversion to belatacept after lung transplantation: report of 10 cases

PONE-D-22-26417R1

Dear Dr. Brugière,

We’re pleased to inform you that your manuscript has been judged scientifically suitable for publication and will be formally accepted for publication once it meets all outstanding technical requirements.

Kind regards,

Niels Olsen Saraiva Câmara, M.D, PhD

Academic Editor

PLOS ONE

Additional Editor Comments (optional):

Reviewers' comments:

Reviewer's Responses to Questions

**Comments to the Author**

1. If the authors have adequately addressed your comments raised in a previous round of review and you feel that this manuscript is now acceptable for publication, you may indicate that here to bypass the “Comments to the Author” section, enter your conflict of interest statement in the “Confidential to Editor” section, and submit your "Accept" recommendation.

Reviewer #1: All comments have been addressed

Reviewer #2: All comments have been addressed

2. Is the manuscript technically sound, and do the data support the conclusions?

Reviewer #1: Yes

Reviewer #2: Partly

3. Has the statistical analysis been performed appropriately and rigorously? 

Reviewer #1: Yes

Reviewer #2: N/A

4. Have the authors made all data underlying the findings in their manuscript fully available?

Reviewer #1: Yes

Reviewer #2: Yes

5. Is the manuscript presented in an intelligible fashion and written in standard English?

Reviewer #1: Yes

Reviewer #2: Yes

6. Review Comments to the Author

Reviewer #1: Thanks for these extended and fair answers to my comments. I think your manuscript will be helpful for transplant community.

Reviewer #2: authors have responded to critiques appropriately. the manuscript is improved and clearer. the data more interpretable and the conclusions supported by the data.

7. PLOS authors have the option to publish the peer review history of their article (what does this mean?). If published, this will include your full peer review and any attached files.

Reviewer #1: No

Reviewer #2: No

---

## [Editor Report · Acceptance letter]

6 Mar 2023

PONE-D-22-26417R1 

Conversion to belatacept after lung transplantation: *report of 10 cases.*

Dear Dr. Brugière:

I'm pleased to inform you that your manuscript has been deemed suitable for publication in PLOS ONE. Congratulations! Your manuscript is now with our production department. 

Kind regards, 

on behalf of

Prof. Niels Olsen Saraiva Câmara 

Academic Editor

PLOS ONE